# Investigation on Physical Properties of IGZO Thin Films under the Conditions of Mixing Oxygen and Argon

Yanyan Zhang [1] and Yong Pan [2,*]

1 School of Pharmacy, Hanzhong Vocational and Technical College, Hanzhong 723000, China
2 College of Science, Xi'an University of Architecture and Technology, Xi'an 710055, China
* Correspondence: panyong@xauat.edu.cn

**Abstract:** Amorphous indium gallium zinc oxide (IGZO) is the most suitable material choice for optoelectronic devices such as thin film transistor (TFT). However, usually, the physical properties of IGZO are of concern rather than the preparation process, which will complicate the control of the main properties of the material. To obtain a simple method of controlling IGZO properties, different proportions of mixed gases of oxygen and argon were added in the process of preparing thin films by pulsed laser deposition (PLD) for studying the fluence of atmosphere on the growth of IGZO. The structure and components of the film are characterized by X-ray diffraction (XRD), which confirmed the amorphous structure. A red-shift of the absorption peak in range of 450–850 nm was generated with the increase in argon concentration. Meanwhile, the transmission spectra showed that the transmittance of the material was lower than 80% in the range of 450–850 nm. Then, different target samples have a wide photoluminescence band at 200–800 nm. Oxygen vacancy defects were considered to be closely related to the photoluminescence behavior. The smallest surface roughness of the films prepared under 50% Ar and 50% $O_2$ and the largest in 100% Ar are proved by atom force microstructure (AFM). Importantly, the greater difference in electrical properties reflects the sensitivity of different oxygen and argon concentrations to material effects. The carrier concentration can be adjusted from $1.08 \times 10^{11}$ to $1.33 \times 10^{16}$ by this method. Finally, the IGZO achieved in this work was used in a transistor, which reflected good diode characteristics.

**Keywords:** thin films; IGZO; oxygen; argon; PLD

## 1. Introduction

InGaZnO (IGZO), a transparent oxide semiconductor, has become the focus of flexible display technology due to many advantages such as high transmittance in the visible region, high mobility of electrical properties, good controllability of carrier concentration, and ability to be easily deposited in a large area and grow at low temperature [1–4]. IGZO has been used in many optoelectronic devices, such as p-n junction rectifier, UV led and transparent FET [5–7]. Amorphous IGZO thin film is a suitable material for a transistor (TFT) with a channel layer, which can meet the high requirements of flexible display.

The basic requirements of channel layer materials for TFT are high electron mobility and high transparency [8,9]. At present, the methods of preparing IGZO films include hydrothermal reaction [10], sol–gel [11] and pulsed laser deposition (PLD) [12]. Among these methods, PLD is a common method for preparing thin films with prominent advantages such as controllable growth conditions, precise control of process parameters and easy realization of directional growth. For thin films prepared by PLD, the substrate temperature, protective pressure in vacuum cavity, distance between target and substrate, laser output power, laser pulse frequency and other process parameters all have influences on the quality of thin films. Only by adjusting or optimizing the preparation parameters can we obtain targeted thin films. However, most of the IGZO films prepared by PLD with one growing parameter controlled do not achieve the desired properties. In the process of

growth, the simultaneous change of multiple parameters should be studied. Meanwhile, most researchers are concerned with the physical properties of materials after mounding rather than in the preparation process, which will make it impossible for us to understand the factors affecting the physical properties and the preparation conditions.

Many researchers are devoted to the preparation and application of IGZO, which is a typical material of amorphous oxide in TFTs and is considered as a recognized material for the channel layer of the next generation of flat panel display TFTs. The amorphous IGZO thin-film transistors (a-IGZO TFTs) meet all the needs of light-emitting diode display, large area liquid crystal and three-dimensional display devices, which are not met by traditional silicon and organic devices. Ref. [13] reported the impact on the electrical properties of the IGZO-TFTs at a fixed $N_2$ gas flow rate on the $SiO_2$ dielectric surface. A high mobility of 33.5 $cm^2$/V s and high carrier concentration was achieved. In Ref. [14], the authors prepared thin films by magnetron sputtering in various oxygen atmospheres at room temperature, and the TFTs were evaluated. Overall, the above studies show that the introduction of $N_2$ and $O_2$ in the preparation of IGZO will have an important impact on its electrical properties. However, the mobility and carrier concentration of the materials mentioned above are still low. Therefore, new changes are needed to improve the electrical properties of materials. To improve the electrical properties of materials, new preparation methods and conditions need to be explored.

Here, we focus on changing the preparation conditions to obtain different electrical properties in terms of IGZO materials. We are committed to studying the properties of IGZO thin film under two different deposition conditions injecting at the same time. Namely, different proportions of mixed gases of oxygen and argon were added in the process of preparing by pulsed laser deposition (PLD) to understand the fluence of atmosphere on growth of IGZO. Then, the properties of structure, morphology, optical and electrical are measured. The novelty of this manuscript can be summarized as new IGZO film preparation conditions for oxygen–argon co-mixing and an ultra-simple method for controlling carrier concentration, mobility and resistivity. This will promote the development of optoelectronic devices such as TFT.

## 2. Materials and Methods

The target of IGZO used in the vacuum chamber was prepared by the method of solid-state reaction. The atom ratio of x($In_2O_3$):y($Ga_2O_3$):z(ZnO) is set as 7:2:1 atom %. High-purity $In_2O_3$ (99.99%), $Ga_2O_3$ (99.999%) and ZnO (99.99%) powders were weighed and pre-sintered for 6 h at 1200 °C after full mixing and grinding. The pre-sintered powder was crushed and ground again; then, it was pressed by hydraulic tablet press. The round cake billet with a diameter of 30 mm and thickness of 5 mm was obtained. The forming pressure is 10 MPa and the holding time is 3 min. The material ratio changed before and after sintering. The concentration $(In_2O_3)_{0.7}(Ga_2O_3)_{0.2}(ZnO)_{0.1}$ was changed to $(In_2O_3)_{0.71}(Ga_2O_3)_{0.13}(ZnO)_{0.12}$. The growth of IGZO thin films was carried out by the PLD method. The laser system used is a Spectra Physics GCR-170 (San Francisco, CA, USA) Nd:YAG pulse laser with a repetition rate of 10 Hz and pulse width of 10 ns. In this experiment, a 355 nm laser beam was used to irradiate the laser beam in PLD. The substrate was quartz glass (10 mm × 10 mm × 0.5 mm), the target base distance was set as 50 mm, the background vacuum was $2.5 \times 10^{-4}$ Pa, the oxygen partial pressure was 1 Pa, and the average power of the laser beam before focusing was 1 Pa. The substrate temperature was room temperature (RT) 20 °C, and the deposition time was 38 min at 450 mW.

The microstructure of thin film was measured by the X-ray Diffraction (XRD) technique (Bruker D8 Advance, Karlsruhe, Germany). The morphology of the films is visualized by AFM (ALPHA 300, WItec, Ulm, Germany). The TFT image is tested by SEM (JEOL JSM 6500F, Tokyo, Japan). The absorption and transmission spectra were tested by a spectrometer (HITACHI U-4100, Tokyo, Japan). The atomic molar ratio of the films was confirmed by X-ray fluorescence spectrometer (PANalytical Magix PW2403, Almelo, The Netherlands). The valence state of IGZO was tested by XPS (X-ray photoelectron spectroscopy using

Escalab 250 Thermo, New York, NY, USA). The electrical properties were tested using a Hall-effect measurement system (Eastchanging ET 9000, Beijing, China).

### 3. Results and Discussion

Figure 1 shows XRD patterns of the IGZO thin films at two atmospheres. There are no sharp peaks in the pattern, illustrating the successful preparation of IGZO in this experiment. The wide peak is found in the ranges of 20–40° (quartz substrate) and 60–80° due to the amorphous structure [15], which is consistent with our purpose in terms of an amorphous thin film. The reason for the amorphous nature is the addition of a grid structure. The reason for using an amorphous state is that the conductivity of single crystal and polycrystal is stronger than that of the amorphous state. The preparation of single crystal is very expensive. The conductivity of polycrystal is uneven, and it is easy to have different brightness between pixels. Amorphous structures are cheap and homogeneous in conductivity; it is enough in industrial application at present [16–18]. The IGZO films with stable amorphous structures do not change significantly upon exposure to two kinds of atmosphere, which indicates that IGZO films with stable amorphous structures have been achieved. It is not accurate to calculate the micro size of amorphous materials with XRD data usually. The first reason is that the signal of the amorphous phase itself will affect the signal of the crystalline phase. Another is that the crystalline phase is generally nanocrystalline, and there is a big error in accurately measuring the grain size by XRD. Moreover, the crystalline phase in the amorphous structure is not very uniform in many cases, and the estimated grain size is easily questioned.

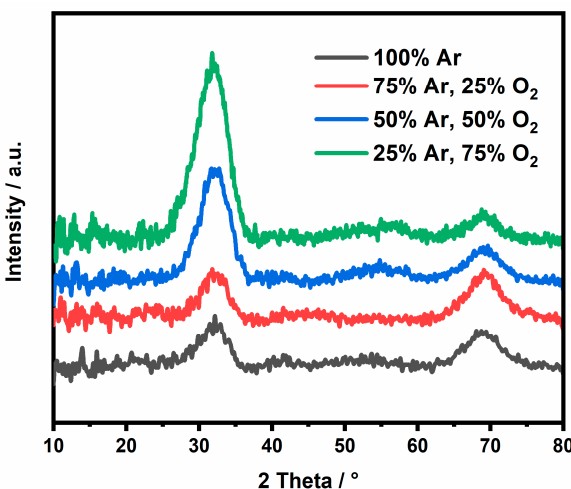

**Figure 1.** The XRD patterns of the IGZO thin films at different deposition atmospheres.

What needs to be explained is that the biggest difference between amorphous materials and crystalline materials is their microstructure layout. Long-range order is the characteristic of a crystalline microstructure, while short-range order and long-range disorder are the characteristics of an amorphous state. Thus, for amorphous IGZO, the simple microstructure parameters just can be calculated in the short-range order. A theoretical model was built for the microstructure of IGZO thin film samples, and theoretical simulation was carried out. According to the lattice structure and the component distribution ratio measured in the experiment, the single cell structure is established, the substitution doping is carried out, and then, the periodic multi cell system is expanded. The theoretical model of a periodic polycrystal system in reciprocal space, which still maintains the indium oxide structure of a body-centered cubic ferromanganese, belongs to the Ia3 space group and the lattice constant a = b = c = 0.876 nm, $\alpha = \beta = \gamma = 109.47°$. The PBE function of the generalized gradient approximation (GGA) is used as the basis function of the super soft pseudopotential plane in the calculation. Only the low kinetic energy with large numerical value is considered in the calculation. The electron exchange interaction is treated as the

generalized gradient approximation. This calculation result is close to that of IGZO in crystal form, but it is not necessarily accurate for IGZO in an amorphous state. H and N are the main impurity elements. According to the XPS test results, their influence on amorphous materials is very small. The main reason is that the total amount of impurities is small and it is not easy to enter the amorphous bonding to affect the absorption and band gap energy [19,20].

In amorphous IGZO, there are many defects, and they are irregular. It is difficult to study the defects of amorphous materials. According to the characteristics of crystalline IGZO, the electronic valence states of indium and gallium atoms conform to the + 3 valence state, and the electronic valence states of zinc atoms conform to the + 2 valence state. There are oxygen defects near each zinc atom, and only oxygen atoms near each zinc atom form Zn-O bonds with them. In amorphous materials, the effect of defects is relatively small [21,22].

The absorption spectrum of the IGZO is shown in Figure 2a. The main absorption in the range of 450–850 nm is confirmed. IGZO films with the highest absorption peaks are prepared in 50% of each atmosphere. The high absorption in the oxygen and argon content is close or equal. Then, the thin film prepared at 75%$-$Ar and 25%$-$O$_2$ also has a higher intensity of absorption. Then, the absorption of the films prepared in 25% Ar atmosphere decreased. The absorption peak intensity of argon decreases, as we only keep argon in the conditions (100% argon). Although the absorption peaks of the films prepared in different deposition atmospheres decreased, the change of absorption spectrum is relatively small. This is consistent with the results of XRD because the microstructure of the material changes slightly. Comparatively speaking, the optical properties of the material are better if the content of argon is higher under the coexistence of the two atmospheres. Meanwhile, the transmission spectrum is shown in Figure 2b. The transmission spectra show that the transmittance of the material is lower than 80% in the range of 450–850 nm, which indicates that there is a certain absorption in this area. The films with the highest transmittance were prepared in 100% Ar atmosphere, which was followed by 25%, 75% and 50%. This is contrary to the trend of absorption spectrum. Clearly, the transmission spectrum is magnified in range of 600–660 nm, which corresponded to the high absorption area. It shows that there is still a broad absorption peak in the amplified transmission spectrum.

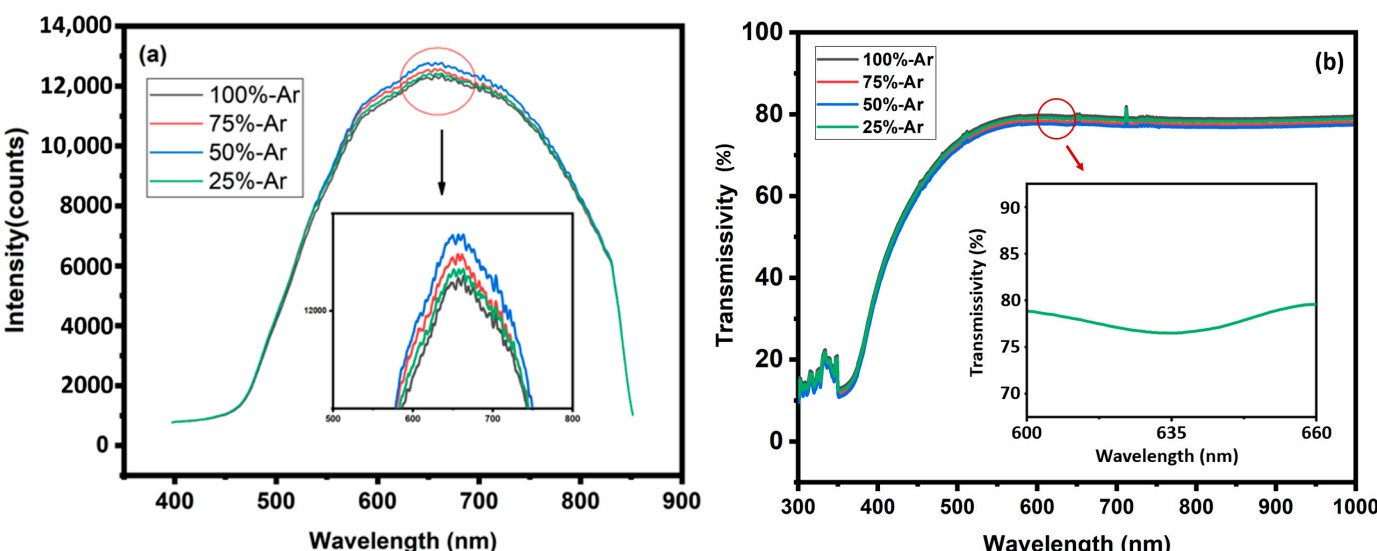

**Figure 2.** The absorption and transmission spectrum of the IGZO at the different deposition atmospheres. (**a**) Absorption spectrum, Inset: the enlarged area of the red part; (**b**) Transmission spectrum, Inset: the enlarged area of the red circular in range of 600–660 nm.

The PL spectrum of IGZO excited by a 325 nm laser is shown in Figure 3. It can be observed that different target samples have a wide luminescence band at 200–800 nm, and the

luminescence center is in the range of 420 nm. Different from the previous speculation, the PL spectrum did not find the emission peak of bulk wurtzite ZnO near 380 nm. Generally speaking, oxygen vacancies in ZnO are considered as donors, and deep level luminescence is closely related to oxygen vacancy defects. For IGZO, oxygen vacancy defects are also considered to be closely related to the photoluminescence behavior. This behavior is caused by the electronic transition modulated by the defect level in the band gap.

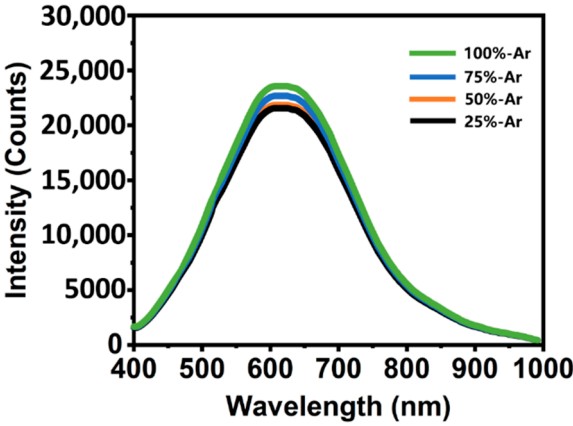

**Figure 3.** The PL spectrum of IGZO at different deposition atmospheres.

To prove the composition of the material, the XPS spectrum was tested. The O 1s peak in IGZO is shown in Figure 4a. According to the standard spectrum, the peak around 531 eV can be considered as the $O_{2-}$ oxide. Then, the peak of In $3d^{5/2}$ is also found in the results. Compared with the standard spectrum, the peak information in this area coincides with the crowning phase of $In_2O_3$ (Figure 4b). Similarly, the peak positions of Ga $2p^{3/2}$ and Ga $2p^{1/2}$ in 1118.5 eV and in 1145.2 eV are marked, respectively. In addition, the binding energy of Ga $2p^{3/2}$ in metallic Ga is 1116.6 eV, so Ga also exists in an oxidation state, which confirmed the existence of $Ga_2O_3$ (Figure 4c). The peaks representing Zn are mainly located at 1022.8 eV and 1022.3 eV, but the binding energies for each film are all above bulk ZnO (1021.7 eV) and metal Zn (1021.1 eV), which illustrates that Zn is in an oxidation state and partially in existence with $Zn^{2+}$ with an anoxic condition.

The AFM image of the IGZO thin film is displayed in the Figure 5. With the change of argon content, the surface morphology of the material did not change dramatically (Figure 5a,b). However, the surface roughness is different from the high peak probability distribution image, which shows that the average peak distribution of the films prepared at 50% Ar and 50% $O_2$ is lower than that of the films prepared at 25% Ar and 75% $O_2$. Meanwhile, the peak distribution of the films prepared at 100% is the highest (the dot line). Therefore, the smallest surface roughness of the films prepared under 50% Ar and 50% $O_2$ is confirmed (Figure 5b). The largest surface roughness prepared under 100% Ar is found (Figure 5c). The reason for these changes can be explained by the structural changes under different preparation conditions. This is consistent with the results of absorption spectra. Meanwhile, the film thickness and RMS surface roughness of the IGZO thin films are displayed in Table 1. The smallest surface roughness of the films prepared under 50% Ar and 50% $O_2$ and largest in 100% Ar are provided.

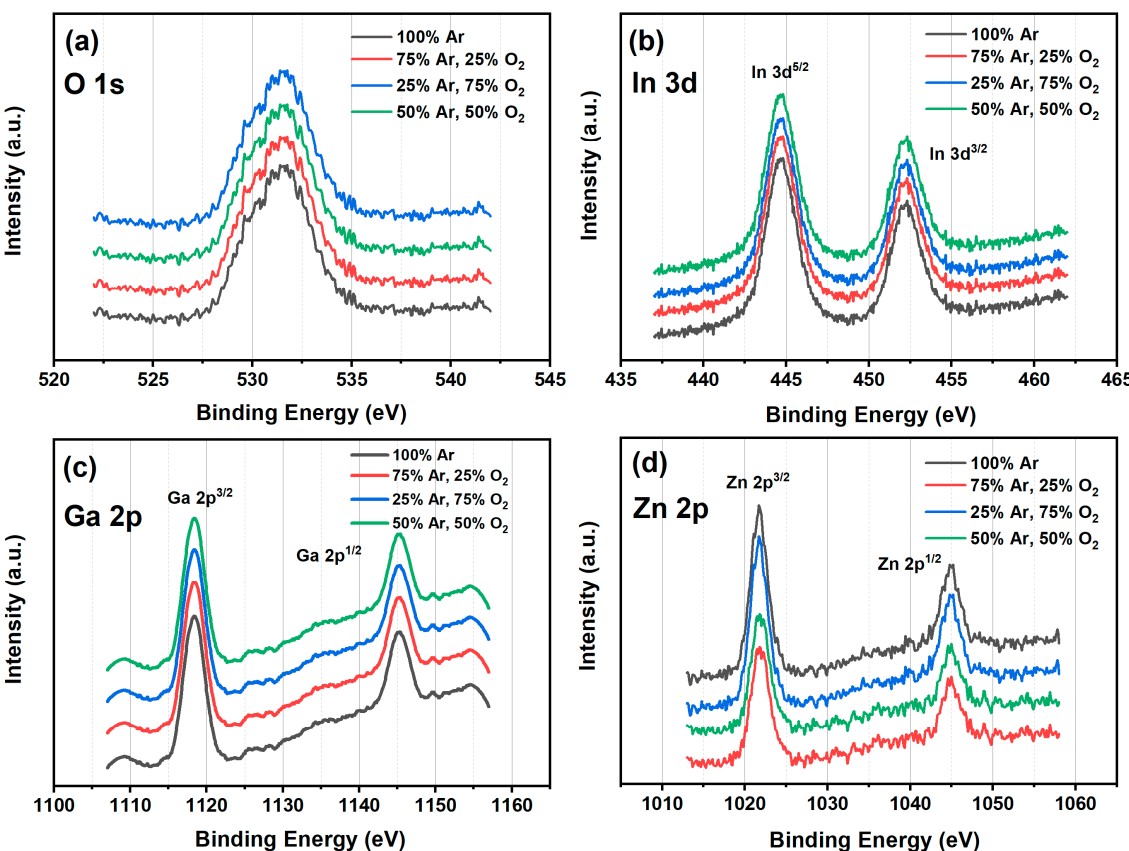

**Figure 4.** The XPS spectrum of the IGZO at different deposition atmospheres. (**a**) O 1s; (**b**) In 3d; (**c**) Ga 2p; (**d**) Zn 2p.

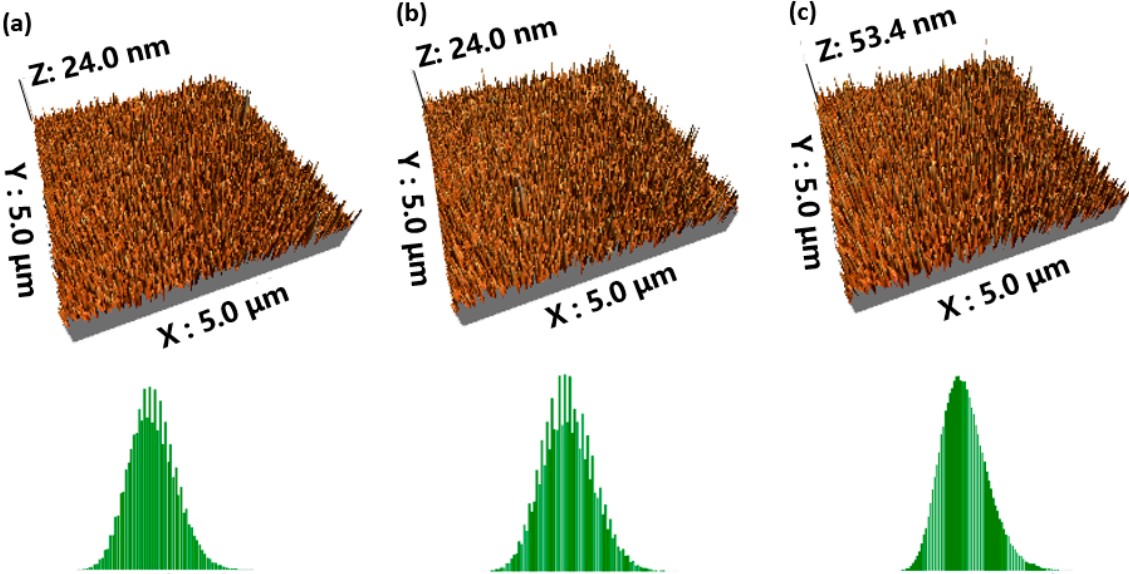

**Figure 5.** The AFM image of the IGZO. (**a**) 25%−Ar 75%−O$_2$; (**b**) 50%−Ar 50%−O$_2$; (**c**) 100%−Ar; The lower row: high peak probability distribution.

**Table 1.** The film thickness and RMS surface roughness of IGZO thin films.

| Conditions | 100% $O_2$ | 50% Ar + 50% $O_2$ | 25% Ar + 75% $O_2$ | 100% Ar |
|---|---|---|---|---|
| Film thickness | 105.48 nm | 82.02 nm | 142.86 nm | 153.93 nm |
| RMS surface roughness | 28.71 nm | 18.94 nm | 22.36 nm | 48.65 nm |

To acknowledge the electrical properties of IGZO thin film, the Hall effect test is conducted in this experiment, as shown in Table 2. The resistance of the material is minimized at the condition of 75% Ar−25% $O_2$. At the same time, the maximum and minimum carrier concentration and mobility are manifest in the film prepared at the condition of 75% Ar−25% $O_2$. The reason for the change can be explained by replacing oxygen with Ar. In fact, the In content in the material we prepared is usually the highest in our previous work [23,24] for improving the Hall mobility. However, when the carrier concentration of a material exceeds 18 orders of magnitude, the material will be transformed into a conductor. Of course, this conductive material is not suitable as a TFT structural layer. Therefore, Ar atmosphere was injected in the chamber to keep their semiconductor properties. The result of $1.33 \times 10^{16}$ carrier concentration confirmed this explanation. In this paper, we introduce new preparation conditions to ensure the semiconductor properties at high $In_2O_3$ content. From the results of the Hall effect, we can conclude that the conductor properties of materials can be changed by our method.

**Table 2.** The result of Hall measurement of IGZO thin film at different deposition atmospheres.

| Conditions | Resistivity (ohm·cm) | Hall Coefficient (cm$^3$·c$^{-1}$) | Carrier Concentration (cm$^{-3}$) | Hall Mobility (cm$^2$·v$^{-1}$·s$^{-1}$) |
|---|---|---|---|---|
| 100% $O_2$ | $5.59 \times 10^{-4}$ | $-1.43 \times 10^{-2}$ | $4.35 \times 10^{20}$ | $2.65 \times 10^1$ |
| 25% Ar + 75% $O_2$ | $2.33 \times 10^4$ | $-6.11 \times 10^6$ | $1.02 \times 10^{12}$ | $2.62 \times 10^2$ |
| 50% Ar + 50% $O_2$ | $1.54 \times 10^3$ | $-3.61 \times 10^4$ | $1.72 \times 10^{14}$ | $2.3 \times 10^1$ |
| 75% Ar + 25% $O_2$ | $8.69 \times 10^1$ | $-4.66 \times 10^2$ | $1.33 \times 10^{16}$ | 5.36 |
| 100% Ar | $1.09 \times 10^4$ | $-5.74 \times 10^7$ | $1.08 \times 10^{11}$ | $5.24 \times 10^3$ |

The results of the I-V test of TFT are shown in Figure 6. Firstly, the structure of the TFT is exhibited in Figure 6a. $SiO_2$ is used as the insulating layer with a thickness of 200 nm. A layer of IGZO was prepared on $SiO_2$ by PLD. The width and length of the channel layer were 40–50 μm and 0.5 mm, respectively. Aluminum is the electrode of this TFT. To show the structure of the TFT more clearly, the cross-section SEM images are presented in Figure 6b. Then, the SEM image of the IGZO TFT is shown in Figure 6c–e. The Al electrode layer deposited on IGZO film is clearly presented by Figure 6c. The size distribution of the electrodes in the range of 40–55 μm is confirmed. Figure 6d exhibited a channel with the size of 45.3 μm. A smooth, uniform channel with less impurities is presented in Figure 6e. The SEM pictures show that TFT devices consisting of IGZO and Al electrodes have been successfully fabricated. From Figure 6f, the device composed of IGZO prepared at 75% Ar and 50% Ar exhibited the better I-V properties due to the obviously forward voltage. This result confirms the conclusions of previous studies such as absorbed properties and morphology. However, the property of the device composed of IGZO prepared at 100% Ar tends to demonstrate a linear change, which does not show a good diode performance. We enlarged some data to show this effect more clearly, as seen in Figure 6g. The cut-off voltage of the composed of IGZO prepared at 75% Ar can be directly found. Meanwhile, the curve of 50% Ar tends to be smoother, which also reflects some diode characteristics.

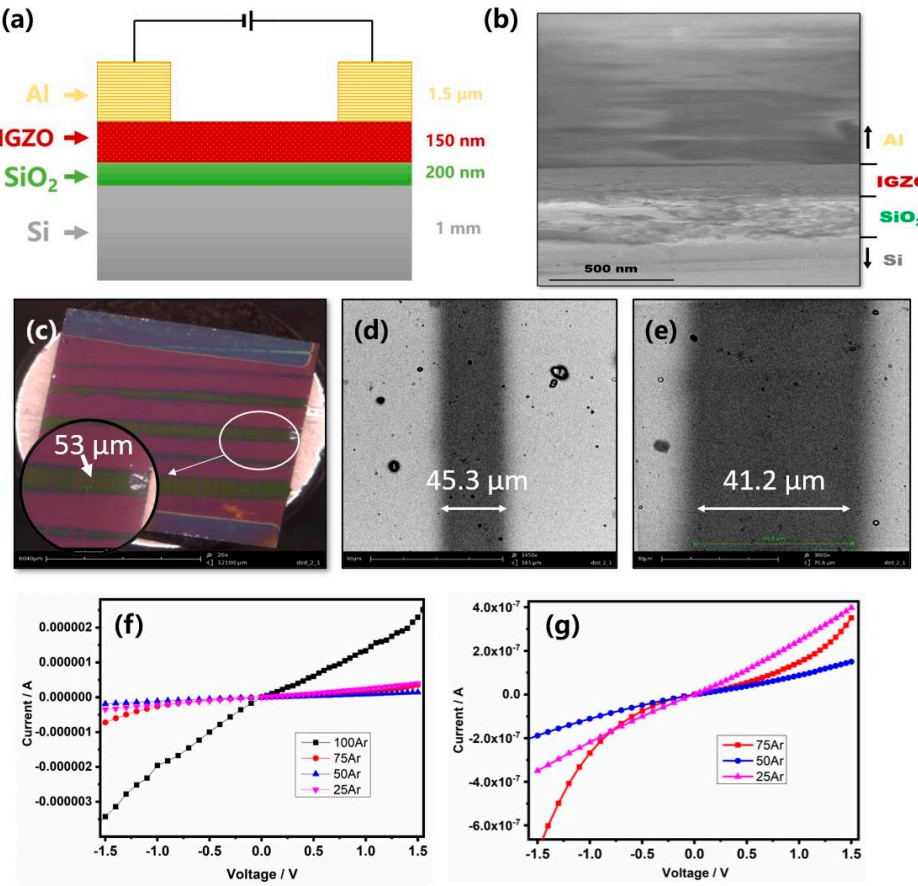

**Figure 6.** The result of I−V test of IGZO and Al TFT. (**a**) The structure of the TFT; (**b**) The cross-section SEM image of TFT. (**c**–**e**) The SEM image of the IGZO TFT at 50%-Ar 50%−O$_2$; (**f**) The I−V date; (**g**) The enlarged local data.

## 4. Conclusions

IGZO thin films were fabricated by the method of PLD at two different deposition atmospheres conditions including Ar and O$_2$. The amorphous structures of all the films were confirmed by XRD. The absorption and transmission spectrum in the range of 450–850 nm was measured. Red-shift of the absorption peak in range of 450–850 nm was generated with the increase in argon concentration. Meanwhile, the transmission spectra showed that the transmittance of the material was lower than 80% in the range of 450–850 nm. The smallest surface roughness of the films prepared under 50% Ar and 50% O$_2$ and the largest in 100% Ar were depicted by the AFM. The Hall effect test illustrated that the material prepared at 75% Ar-25% O$_2$ was very suitable for the TFT channel layer. The TFT device consisting of IGZO and Al electrodes had been successfully fabricated and imaged by SEM. The device composed of IGZO prepared at 75% Ar and 50% Ar displayed the better I−V properties due to the obviously forward voltage. All the results showed that the optical and electrical properties were better if the content of argon is higher under the coexistence of the two atmospheres, especially in the environment of 75% Ar-25% O$_2$. In conclusion, the conductor properties of materials can be changed by our method, especially in electrical properties. The resistivity, carrier concentration and Hall mobility can be adjusted from $8.69 \times 10^{-4}$ to $2.33 \times 10^4$, $1.08 \times 10^{11}$ to $1.33 \times 10^{16}$ and $5.36$ to $5.24 \times 10^3$. This change will help to improve the performance of TFT devices.

**Author Contributions:** Formal analysis, Y.Z.; Funding acquisition, Y.P.; Investigation, Y.Z.; Software, Y.Z.; Writing—original draft, Y.P. All authors have read and agreed to the published version of the manuscript.

**Funding:** Natural Science Foundation of Shaanxi Province (2022JQ-652). The project of talent introduction of Xi'an University of architecture and technology (1960320034).

**Institutional Review Board Statement:** Not applicable.

**Informed Consent Statement:** Not applicable.

**Data Availability Statement:** The data presented in this study are available on request from the corresponding author.

**Conflicts of Interest:** The authors declare no conflict of interest.

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
