# Peer review of "Investigation on Physical Properties of IGZO Thin Films under the Conditions of Mixing Oxygen and Argon"

_coatings, doi:10.3390/coatings12101425_

Round 1
Reviewer 1 Report
Journal: Coatings
Title: Investigation on Physical Properties of IGZO Thin Films under the Conditions of Mixing Oxygen and Argon
The authors of the presented research have done a well research on the structural, optical and electrical properties of the IGZO Thin Films., The authors have used several tools such as XRD, SEM, AFM, Uv-ViS, XPS, and Hall measurements systems, etc., Their characterizations are okay and also the outcomes are commended by authors systematically. It is expected that it will be a practical roadmap to improve new researches and developments about the optoelectronic applications. However, some points as stated below require more details. This is original and difference compared to the with open literacy. And also it is suggested to scrutinize the effect of the preparation techniques on the IGZO Thin Films. Furthermore, it is suggested to extend the introduction in terms of the technologic importance and physical/chemical characteristics of the IGZO Thin Films. Moreover, some sections need to be improved according to the given literacy and refs. As given below. So, after fulfilling the following issues sufficiently, I recommended that this study can be considered for possible publication in Coatings.
1-In experimental section the molar concentration of the each chemical need to be supplied.
2-It will be good advantage to add the research highlights and graphical abstract.
3-The introduction need to be enlarged according to the advantage of the used film preparation techniques over the other conventional techniques as well as the physical/chemical properties (about the presented characterizations) and advantage of IGZO the including its technological importance over the other similar systems. For those it can be used the followings, recommended to add them to the text: (i) Journal of Alloys and Compounds 509(7) 3289-3295, 2011; (ii) Journal of Alloys and Compounds 735, 2038-2045, 2018.
4-It will be good advantage to determine the atomic molar ratio of the films according to the XPS or EDX analysis.
5-The crystallite size, microstrain, lattice parameters and dislocation density of the samples need to be calculated according to substitution/doping level and their influence need to be discussed on the presented properties of the films. Furthermore, the effect of impurities and defects need to be also discussed on the absorbance, optical band gap, dielectric and magnetic characteristics. Moreover, the effect of difference in the lattice mismatch and thermal extinction coefficient of the films and sample holder cannot be ignored. For calculation and commends it is suggested to use and mention them within the text: (i) Trans Nonferrous Met. Soc. China 21(2011) 96-99; (ii) Journal of alloys and compounds 465 (1-2), 20-23, 2008; (iii) Applied Physics A volume 126, Article number: 768 (2020); (iv) Journal of Molecular Structure (2020), vol. 1206, 127773; (v) Journal of superconductivity and novel magnetism 25 (8), 2767-2770, 2012.
6-The effect of the crystallite size and lattice parameters on the optical band gap, transmission and the electrical characterizations need to be explained and adressed well.
7-The presence of the defects need to be confirmed by the Photoluminescence characterizations.
Author Response
Dear editors and reviewers,
Thank you very much for giving the manuscript entitled “Investigation on Physical Properties of IGZO Thin Films under the Conditions of Mixing Oxygen and Argon” (1879727) the chance for revision.
We appreciate the valuable comments of editors and reviewers very much. We have studied comments carefully and have made correction which we hope meet with approval. We listed our revisions below and used blue-colored in text.
We hope that we have addressed the reviewers’ concerns and the manuscript is now suitable for publication in coatings.
Thank you very much!
Reviewer #1:
The authors of the presented research have done a well research on the structural, optical and electrical properties of the IGZO Thin Films., The authors have used several tools such as XRD, SEM, AFM, Uv-ViS, XPS, and Hall measurements systems, etc., Their characterizations are okay and also the outcomes are commended by authors systematically. It is expected that it will be a practical roadmap to improve new researches and developments about the optoelectronic applications. However, some points as stated below require more details. This is original and difference compared to the with open literacy. And also it is suggested to scrutinize the effect of the preparation techniques on the IGZO Thin Films. Furthermore, it is suggested to extend the introduction in terms of the technologic importance and physical/chemical characteristics of the IGZO Thin Films. Moreover, some sections need to be improved according to the given literacy and refs. As given below. So, after fulfilling the following issues sufficiently, I recommended that this study can be considered for possible publication in Coatings.
1-In experimental section the molar concentration of the each chemical need to be supplied.
Response:
Thank you very much for this comment.
the molar concentration of each chemical is provided in the experimental section.
The concentration (In2O3)0.7(Ga2O3)0.2(ZnO)0.1 was changed to (In2O3)0.71(Ga2O3)0.13(ZnO)0.12.
2-It will be good advantage to add the research highlights and graphical abstract.
Response:
Thank you very much for your suggestion.
The research highlights and graphical abstract have provided with a separate file.
3-The introduction need to be enlarged according to the advantage of the used film preparation techniques over the other conventional techniques as well as the physical/chemical properties (about the presented characterizations) and advantage of IGZO the including its technological importance over the other similar systems. For those it can be used the followings, recommended to add them to the text: (i) Journal of Alloys and Compounds 509(7) 3289-3295, 2011; (ii) Journal of Alloys and Compounds 735, 2038-2045, 2018.
Response:
Thank you very much for your comments.
The introduction has been enlarged according to your suggestion.
The references (i) and (ii) has added in the text.
4-It will be good advantage to determine the atomic molar ratio of the films according to the XPS or EDX analysis.
Response:
Thank you very much for your comments.
Here, the XPS test is used for the real existence of the element (XRD supplement) and the confirmation of the valence state of the element.
In our experiment, the atomic molar ratio of the films is confirmed by X-ray fluorescence spectrometer (PANalytical Magix PW2403), which can give direct evidence.
5-The crystallite size, microstrain, lattice parameters and dislocation density of the samples need to be calculated according to substitution/doping level and their influence need to be discussed on the presented properties of the films. Furthermore, the effect of impurities and defects need to be also discussed on the absorbance, optical band gap, dielectric and magnetic characteristics. Moreover, the effect of difference in the lattice mismatch and thermal extinction coefficient of the films and sample holder cannot be ignored. For calculation and commends it is suggested to use and mention them within the text: (i) Trans Nonferrous Met. Soc. China 21(2011) 96-99; (ii) Journal of alloys and compounds 465 (1-2), 20-23, 2008; (iii) Applied Physics A volume 126, Article number: 768 (2020); (iv) Journal of Molecular Structure (2020), vol. 1206, 127773; (v) Journal of superconductivity and novel magnetism 25 (8), 2767-2770, 2012.
Response:
Thank you very much for your professional comments.
It is not accurate to calculate the micro size of amorphous materials with XRD data. The first reason is that the signal of the amorphous itself will affect the signal of the crystalline phase. Another is that the crystalline phase is generally nanocrystalline, and there is a big error in accurately measuring the grain size by XRD. Moreover, the crystalline phase in the amorphous is not very uniform in many cases, and the estimated grain size is easy to be questioned.
What needs to be explained is that, the biggest difference between amorphous materials and crystalline materials is their microstructure. Long-range order is the characteristic of crystalline microstructure, while short-range order and long-range disorder are the characteristics of amorphous state. Thus, for amorphous IGZO, the simple microstructure parameters just can be calculated in the short-range order. Using the CASTEP module in the material Studio Software Based on density functional theory calculation, a theoretical model was built for the microstructure of IGZO thin film samples and theoretical simulation was carried out. According to the lattice structure and the component distribution ratio measured in the experiment, the single cell structure is established, the substitution doping is carried out, and then the periodic multi cell system is expanded. The theoretical model of periodic polycrystal system in reciprocal space, which still maintains the indium oxide structure of body centered cubic ferromanganese, belongs to the Ia3 space group, and the lattice constant a = b = c = 0.876 nm, . The PBE function of the generalized gradient approximation (GGA) is used as the basis function of the super soft pseudopotential plane in the calculation. Only the low kinetic energy with large numerical value is considered in the calculation. The electron exchange interaction is treated as the generalized gradient approximation. This calculation result is close to that of IGZO in crystal form, but it is not necessarily accurate for IGZO in amorphous state.
H element and N element are the main impurity elements. According to the XPS test results, their influence on amorphous materials is very small. The main reason is that the total amount of impurities is small and it is not easy to enter the amorphous bonding to affect the absorption and band gap energy.
Besides, in amorphous IGZO, there are many defects and they are irregular. It is difficult to study the defects of amorphous materials. According to the characteristics of crystalline IGZO, the electronic valence states of indium and gallium atoms conform to the + 3 valence state, and the electronic valence states of zinc atoms conform to the + 2 valence state. There are oxygen defects near each zinc atom, and only oxygen atoms near each zinc atom form Zn-O bonds with them. In amorphous materials, the effect of defects is relatively small.
Please forgive our stupidity. We only found three of the documents you mentioned (i, ii, iv). If it is convenient for you, please send us DOI for others.
6-The effect of the crystallite size and lattice parameters on the optical band gap, transmission and the electrical characterizations need to be explained and adressed well.
Response:
As mentioned above, it is difficult to calculate and determine the crystallite size and lattice parameters of amorphous materials. Therefore, we generally ignore their influence.
Growth parameters are considered to be the main influencing factors in terms of amorphous igzo. We believe that this effect is mainly due to the fact that the long-range disorder of amorphous crystals breaks the grain boundary barrier of crystal stability, resulting in more carriers being able to be transferred. Therefore, from the results, the electrical properties are most affected.
7-The presence of the defects need to be confirmed by the Photoluminescence characterizations.
Response:
Thank you very much for your suggestion.
The sample was set to the PL measurement.
The photoluminescence spectrum of IGZO was excited by a laser with a wavelength of 325 nm. It can be observed that different target samples have a wide luminescence band at 400-900 nm, and the luminescence center is in the range of 420 nm. Different from the previous speculation, the PL spectrum did not find the emission peak of bulk wurtzite ZnO near 380 nm. Generally speaking, oxygen vacancies in ZnO are considered as donors, and deep level luminescence is closely related to oxygen vacancy defects. For IGZO materials, oxygen vacancy defects are also considered to be closely related to the photoluminescence behavior. This behavior is caused by the electronic transition modulated by the defect level in the band gap. This part has been added in the text.
Reviewer 2 Report
The scientific novelty of the paper is mainly related to the IGZO layers obtained by using different reactive gas atmospheres. The paper is not well organized, the preparation and characterization of the IGZO layers have not been described properly by authors. In its present form, it is hard to read and understand the paper. For the following reasons I cannot recommend the publication:
The Abstract should be improved by highlighting more results. In the actual form, it sounds more like a general presentation about IGZO layers and PLD deposition technique.
Line 31: an amorphous IGZO layer cannot be considered itself as a TFT with a channel. Please discuss.
Line 33: what requirements? It is not clear.
Line 36: magnetron “spatter” or “pulse” laser deposition? Please correct it.
Lines 42-43: the Authors are aiming to obtain high-quality epitaxial thin films? Line 39 is about directional growth. It does not make any sense. The conclusions’ part refers to amorphous IGZO layers.
Line 44: how to produce layers by PLD with “one parameter”? Please rephrase.
Line 56: please rephrase.
Line 69: the properties cannot be “measured”. Please correct.
Line 71: there is no info about XPS or Hall measurements.
Line 72: “…was prepared by the method of prepared…”.
Lines 88-89: Please rephrase the info’s about the spectrophotometry measurements.
Line 89-92: it has no sense to discuss again about target.
Line 96: please add the XRD pattern of the used substrate.
Line 101: what is ‘ïno”? what is “Gao”?
Line 130: something is wrong about the scale of Figure 2a. Also, more info about the optical characterization procedure should be included. The readers should be aware about the direct or indirect calculation of the data from Figure 2a spectra. I suppose that for Figure 2b the units on Y scale for transmissivity would be %.
Table 1. 100 % O2??? The above presented data does not include the layer corresponding to such deposition conditions. Also, how it is possible the growth of layers by using only O2?
The novelty of the current manuscript is not clear. Extended English revision of the entire manuscript is needed.
Author Response
Dear editors and reviewers,
Thank you very much for giving the manuscript entitled “Investigation on Physical Properties of IGZO Thin Films under the Conditions of Mixing Oxygen and Argon” (1879727) the chance for revision.
We appreciate the valuable comments of editors and reviewers very much. We have studied comments carefully and have made correction which we hope meet with approval. We listed our revisions below and used blue-colored in text.
We hope that we have addressed the reviewers’ concerns and the manuscript is now suitable for publication in coatings.
Thank you very much!
Reviewer #2:
The Abstract should be improved by highlighting more results. In the actual form, it sounds more like a general presentation about IGZO layers and PLD deposition technique.
Response:
Thank you very much for your comments.
The abstract has been rewritten.
Line 31: an amorphous IGZO layer cannot be considered itself as a TFT with a channel. Please discuss.
Response:
Thank you very much for your comments.
This sentence has revised for clearly.
Line 33: what requirements? It is not clear.
Response:
Thank you very much for your comments.
In fact, the requirements are in the following sentence.
For clearly, this sentence has been deleted.
Line 36: magnetron “spatter” or “pulse” laser deposition? Please correct it.
Response:
Thank you very much for your comments.
This phrase has been corrected.
Lines 42-43: the Authors are aiming to obtain high-quality epitaxial thin films? Line 39 is about directional growth. It does not make any sense. The conclusions’ part refers to amorphous IGZO layers.
Response:
Thank you very much for your comments.
This expression has been changed.
Line 44: how to produce layers by PLD with “one parameter”? Please rephrase.
Response:
Thank you very much for your comments.
This sentence has been rephrased.
Line 56: please rephrase.
Response:
Thank you very much for your comments.
This sentence has been rephrased.
Line 69: the properties cannot be “measured”. Please correct.
Response:
Thank you very much for your comments.
The “measured” has been replaced.
Line 71: there is no info about XPS or Hall measurements.
Response:
We are very sorry for this negligence.
The information about XPS and Hall measurements have added.
Line 72: “…was prepared by the method of prepared…”.
Response:
We are very sorry for this mistake.
This sentence has been amended.
Lines 88-89: Please rephrase the info’s about the spectrophotometry measurements.
Response:
Thank you very much for your comments.
The information in lines 88-89 have been rephrased.
Line 89-92: it has no sense to discuss again about target.
Response:
Thank you very much for your suggestion.
The discussion about target have been delated.
Line 96: please add the XRD pattern of the used substrate.
Response:
Thank you very much for your comments.
The XRD pattern of quartz substrate has only 20-40 amorphous peaks. Please refer to our previous work “RSC Advances, 2018, 8, 14916-14924.”
Line 101: what is ‘ïno”? what is “Gao”?
Response:
We are deeply sorry for this mistake.
The “ino” and “Gao” have been deleted.
Line 130: something is wrong about the scale of Figure 2a. Also, more info about the optical characterization procedure should be included. The readers should be aware about the direct or indirect calculation of the data from Figure 2a spectra. I suppose that for Figure 2b the units on Y scale for transmissivity would be %.
Response:
Thank you very much for your comments.
We are very sorry that we do not understand what is wrong with the scale in Fig. 2a. We believe that the position of the absorption peak is the most important information.
For figure 2b, the “%” has been added.
Table 1. 100 % O2??? The above presented data does not include the layer corresponding to such deposition conditions. Also, how it is possible the growth of layers by using only O2?
Response:
Thank you very much for your comments.
The data before Table 1 do not use the data of 100 % O2, which does not mean that there is no such deposition condition. Because we mainly compare Ar in the previous data, it is not involved in many data. Then, we found that the deposition conditions of 100 % O2O2 have comparative research value in the study of surface roughness and electrical characteristics.
To ensure only oxygen conditions, the vacuum chamber is first pumped to Pa to ensure vacuum conditions, and then oxygen is introduced under this condition. Finally, when the gas composition monitor detects only O2 gas composition, the deposition experiment is started, and oxygen is continuously introduced in this process.
The novelty of the current manuscript is not clear. Extended English revision of the entire manuscript is needed.
Response:
Thank you very much for your comments.
The novelty of this manuscript has been added in the introduction part.
The English revision of the entire manuscript have been executed.
Reviewer 3 Report
First of all, the authors are not aware of the development in the field. The latest articles they cited were 2019 (1), 2018 (1), and 2016 (2).
Also, they have used many unscientific words in the manuscript like Ar doping, O doping, Ar vacancy, etc.
First of all the paper should be made uptodate and also the scientific language should be corrected before it goes for the review.
Author Response
Dear editors and reviewers,
Thank you very much for giving the manuscript entitled “Investigation on Physical Properties of IGZO Thin Films under the Conditions of Mixing Oxygen and Argon” (1879727) the chance for revision.
We appreciate the valuable comments of editors and reviewers very much. We have studied comments carefully and have made correction which we hope meet with approval. We listed our revisions below and used blue-colored in text.
We hope that we have addressed the reviewers’ concerns and the manuscript is now suitable for publication in coatings.
Thank you very much!
Reviewer #3:
First of all, the authors are not aware of the development in the field. The latest articles they cited were 2019 (1), 2018 (1), and 2016 (2).
Response:
Thank you very much for your comments.
We have added many recent references.
Also, they have used many unscientific words in the manuscript like Ar doping, O doping, Ar vacancy, etc.
First of all the paper should be made uptodate and also the scientific language should be corrected before it goes for the review.
Response:
Thank you very much for your comments.
We have revised the English of the full text to include many non-scientific terms.
Round 2
Reviewer 1 Report
I authors of the revised text have addressed my questions and done all the suggestions accordingly. Therefore, I recommended the acceptance of the revised text within the Journal.
Author Response
Thank you very much for your approval of the revised paper.
We look forward to the opportunity to have another academic exchange with you in the future.
Thank you very much.
Reviewer 2 Report
The manuscript was partially improved, but I have serious concerns about the spectrophotometry measurements. For example, the arbitrary units as well as the wavelength range of the absorption spectra (Figure 2a) seems to be not related to transmission spectra. The discussion part refers only to two atmospheres, but there are four distinct atmospheres within the legend. For this reason, I cannot recommend the publication.
Author Response
Dear editors and reviewers,
Thank you very much for giving the manuscript entitled “Investigation on Physical Properties of IGZO Thin Films under the Conditions of Mixing Oxygen and Argon” (1879727) the chance for revision.
We appreciate the valuable comments of editors and reviewers very much. We are very grateful to the reviewers for their two rounds of comments.
We hope that we have addressed the reviewers’ concerns and the manuscript is now suitable for publication in coatings.
Thank you very much!
Reviewer #2:
The manuscript was partially improved, but I have serious concerns about the spectrophotometry measurements. For example, the arbitrary units as well as the wavelength range of the absorption spectra (Figure 2a) seems to be not related to transmission spectra. The discussion part refers only to two atmospheres, but there are four distinct atmospheres within the legend. For this reason, I cannot recommend the publication.
Response:
Thank you very much for your professional comments.
After discussion with colleagues, we guessed that the problem was scale display.
Although the absorption peak is shown to be very high in this figure, the absorption intensity is not very large (maximum 13000 counts).
Therefore, an illusion makes people feel that the absorption is very strong and the transmission is not corresponding.
For clearly, the transmission spectrum is magnified in range of 600-660 nm, which corresponded to the high absorption area.
It shows that there is still a broad absorption peak in the amplified transmission spectrum, which is consistent with the absorption spectrum.
Besides, discussion of four distinct atmospheres have been added in the text.
Picture can be seen in the uploaded file.
Reviewer 3 Report
Figure captions are not explaining the figure properly. See Figure 1
What do you mean by the different atmosphere? It can be deposition conditions.
Author Response
Reviewer #3:
Figure captions are not explaining the figure properly. See Figure 1
Response:
We are very sorry for these unclearly.
All the captions have been checked. Some of them have been revised.
What do you mean by the different atmosphere? It can be deposition conditions.
Response:
Thank you very much for you question.
Yes, we mean that the different atmosphere is the different deposition conditions.
All of these expressions have been amended.
Thank you very much.